# Estimating Accuracy from Unlabeled Data:
# A Probabilistic Logic Approach

**Emmanouil A. Platanios**
Carnegie Mellon University
Pittsburgh, PA
e.a.platanios@cs.cmu.edu

**Hoifung Poon**
Microsoft Research
Redmond, WA
hoifung@microsoft.com

**Tom M. Mitchell**
Carnegie Mellon University
Pittsburgh, PA
tom.mitchell@cs.cmu.edu

**Eric Horvitz**
Microsoft Research
Redmond, WA
horvitz@microsoft.com

## Abstract

We propose an efficient method to estimate the accuracy of classifiers using only unlabeled data. We consider a setting with multiple classification problems where the target classes may be tied together through logical constraints. For example, a set of classes may be mutually exclusive, meaning that a data instance can belong to at most one of them. The proposed method is based on the intuition that: (i) when classifiers agree, they are more likely to be correct, and (ii) when the classifiers make a prediction that violates the constraints, at least one classifier must be making an error. Experiments on four real-world data sets produce accuracy estimates within a few percent of the true accuracy, using solely unlabeled data. Our models also outperform existing state-of-the-art solutions in both estimating accuracies, and combining multiple classifier outputs. The results emphasize the utility of logical constraints in estimating accuracy, thus validating our intuition.

## 1 Introduction

Estimating the accuracy of classifiers is central to machine learning and many other fields. Accuracy is defined as the probability of a system's output agreeing with the true underlying output, and thus is a measure of the system's performance. Most existing approaches to estimating accuracy are *supervised*, meaning that a set of labeled examples is required for the estimation. Being able to estimate the accuracies of classifiers using only unlabeled data is important for many applications, including: (i) any *autonomous learning system* that operates under no supervision, as well as (ii) *crowdsourcing* applications, where multiple workers provide answers to questions, for which the correct answer is unknown. Furthermore, tasks which involve making several predictions which are tied together by *logical constraints* are abundant in machine learning. As an example, we may have two classifiers in the Never Ending Language Learning (NELL) project [Mitchell et al., 2015] which predict whether noun phrases represent animals or cities, respectively, and we know that something cannot be both an animal and a city (i.e., the two categories are mutually exclusive). In such cases, it is not hard to observe that if the predictions of the system violate at least one of the constraints, then at least one of the system's components must be wrong. This paper extends this intuition and presents an *unsupervised* approach (i.e., only *unlabeled data* are needed) for estimating accuracies that is able to use information provided by such logical constraints. Furthermore, the proposed approach is also able to use any available labeled data, thus also being applicable to *semi-supervised* settings.

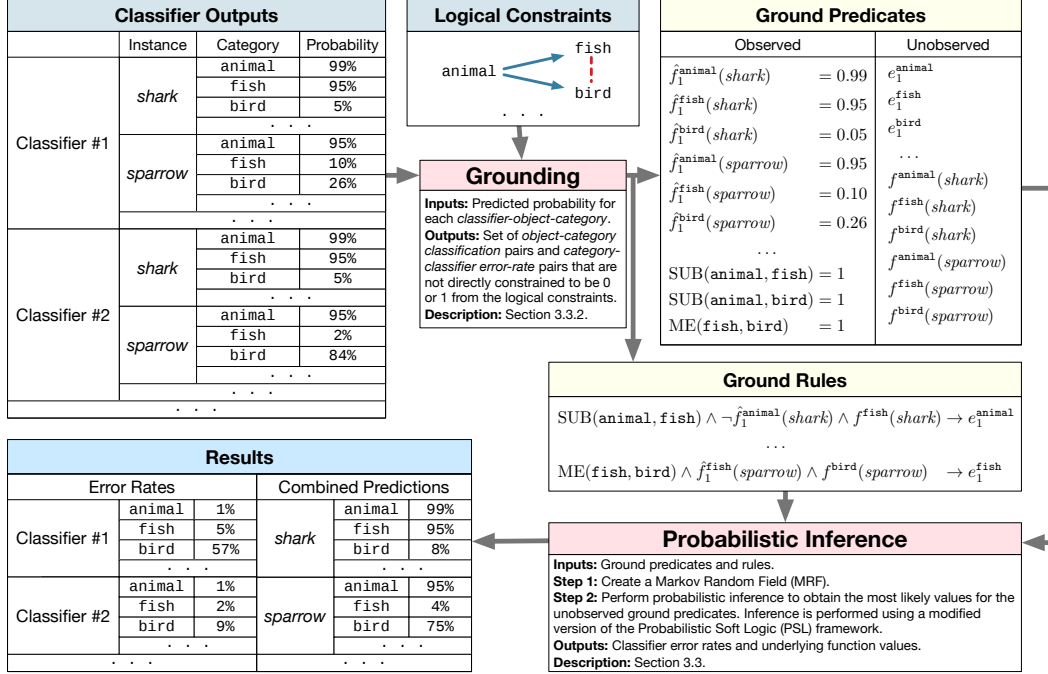

Figure 1: System overview diagram. The classifier outputs (corresponding to the function approximation outputs) and the logical constraints make up the system inputs. The representation of the logical constraints in terms of the function approximation error rates is described in section 3.2. In the logical constraints box, blue arrows represent subsumption constraints, and labels connected by a red dashed line represent a mutually exclusive set. Given the inputs, the first step is grounding (computing all feasible ground predicates and rules that the system will need to perform inference over) and is described in section 3.3.2. In the ground rules box, $\wedge$, $\neg$, $\rightarrow$ correspond to the logic AND, OR, and IMPLIES. Then, inference is performed in order to infer the most likely truth values of the unobserved ground predicates, given the observed ones and the ground rules (described in detail in section 3.3). The results constitute the outputs of our system and they include: (i) the estimated error rates, and (ii) the most likely target function outputs (i.e., combined predictions).

We consider a "multiple approximations" problem setting in which we have several different approximations, $\hat{f}_1^d, \dots, \hat{f}_{N^d}^d$, to a set of target boolean classification functions, $f^d : \mathcal{X} \mapsto \{0, 1\}$ for $d = 1, \dots, D$, and we wish to know the true accuracies of each of these different approximations, using only unlabeled data, as well as the response of the true underlying functions, $f^d$. Each value of $d$ characterizes a different *domain* (or problem setting) and each domain can be interpreted as a class or category of objects. Similarly, the function approximations can be interpreted as classifying inputs as belonging or not to these categories. We consider the case where we may have a set of logical constraints defined over the domains. Note that, in contrast with related work, we allow the function approximations to provide soft responses in the interval $[0, 1]$ (as opposed to only allowing binary responses — i.e., they can now return the probability for the response being 1), thus allowing modeling of their "certainty". As an example of this setting, to which we will often refer throughout this paper, let us consider a part of NELL, where the input space of our functions, $\mathcal{X}$, is the space of all possible noun phrases (NPs). Each target function, $f^d$, returns a boolean value indicating whether the input NP belongs to a category, such as "city" or "animal", and these categories correspond to our domains. There also exist logical constraints between these categories that may be hard (i.e., strongly enforced) or soft (i.e., enforced in a probabilistic manner). For example, "city" and "animal" may be mutually exclusive (i.e., if an object belongs to "city", then it is unlikely that it also belongs to "animal"). In this case, the function approximations correspond to different classifiers (potentially using a different set of features / different views of the input data), which may return a probability for a NP belonging to a class, instead of a binary value. Our goal is to estimate the accuracies of these classifiers using only unlabeled data. In order to quantify accuracy, we define the error rate of classifier $j$ in domain $d$ as $e_j^d \triangleq \mathbb{P}_{\mathcal{D}}[\hat{f}_j^d(X) \neq f^d(X)]$, for the binary case, for $j = 1, \dots, N^d$, where

$\mathcal{D}$ is the true underlying distribution of the input data. Note that accuracy is equal to one minus error rate. This definition may be relaxed for the case where $\hat{f}_j^d(X) \in [0, 1]$ representing a probability: $e_j^d \triangleq \hat{f}_j^d(X)\mathbb{P}_\mathcal{D}[f^d(X)\neq 1] + (1 - \hat{f}_j^d(X))\mathbb{P}_\mathcal{D}[f^d(X)\neq 0]$, which resembles an expected probability of error. Even though our work is motivated by the use of logical constraints defined over the domains, we also consider the setting where there are no such constraints.

## 2  Related Work

The literature covers many projects related to estimating accuracy from unlabeled data. The setting we are considering was previously explored by Collins and Singer [1999], Dasgupta et al. [2001], Bengio and Chapados [2003], Madani et al. [2004], Schuurmans et al. [2006], Balcan et al. [2013], and Parisi et al. [2014], among others. Most of their approaches made some strong assumptions, such as assuming independence given the outputs, or assuming knowledge of the true distribution of the outputs. None of the previous approaches incorporated knowledge in the form of logical constraints. Collins and Huynh [2014] review many methods that were proposed for estimating the accuracy of medical tests in the absence of a gold standard. This is effectively the same problem that we are considering, applied to the domains of medicine and biostatistics. They present a method for estimating the accuracy of tests, where these tests are applied in multiple different populations (i.e., different input data), while assuming that the accuracies of the tests are the same across the populations, and that the test results are independent conditional on the true "output". These are similar assumptions to the ones made by several of the other papers already mentioned, but the idea of applying the tests to multiple populations is new and interesting. Platanios et al. [2014] proposed a method relaxing some of these assumptions. They formulated the problem of estimating the error rates of several approximations to a function as an optimization problem that uses agreement rates of these approximations over unlabeled data. Dawid and Skene [1979] were the first to formulate the problem in terms of a graphical model and Moreno et al. [2015] proposed a nonparametric extension to that model applied to crowdsourcing. Tian and Zhu [2015] proposed an interesting max-margin majority voting scheme for combining classifier outputs, also applied to crowdsourcing. However, all of these approaches were outperformed by the models of Platanios et al. [2016], which are most similar to the work of Dawid and Skene [1979] and Moreno et al. [2015]. To the best of our knowledge, our work is the first to use logic for estimating accuracy from unlabeled data and, as shown in our experiments, outperforms all competing methods. Logical constraints provide additional information to the estimation method and this partially explains the performance boost.

## 3  Proposed Method

Our method consists of: (i) defining a set of logic rules for modeling the logical constraints between the $f^d$ and the $\hat{f}_j^d$, in terms of the error rates $e_j^d$ and the known logical constraints, and (ii) performing probabilistic inference using these rules as priors, in order to obtain the most likely values of the $e_j^d$ and the $f^d$, which are not observed. The intuition behind the method is that if the constraints are violated for the function approximation outputs, then at least one of these functions has to be making an error. For example, in the NELL case, if two function approximations respond that a NP belongs to the "city" and the "animal" categories, respectively, then at least one of them has to be making an error. We define the form of the logic rules in section 3.2 and then describe how to perform probabilistic inference over them in section 3.3. An overview of our system is shown in figure 1. In the next section we introduce the notion of *probabilistic logic*, which fuses classical logic with probabilistic reasoning and that forms the backbone of our method.

### 3.1  Probabilistic Logic

In *classical logic*, we have a set of *predicates* (e.g., mammal($x$) indicating whether $x$ is a mammal, where $x$ is a variable) and a set of *rules* defined in terms of these predicates (e.g., mammal($x$) $\rightarrow$ animal($x$), where "$\rightarrow$" can be interpreted as "implies"). We refer to predicates and rules defined for a particular instantiation of their variables as *ground predicates* and *ground rules*, respectively (e.g., mammal(whale) and mammal(whale) $\rightarrow$ animal(whale)). These ground predicates and rules take boolean values (i.e., are either true or false — for rules, the value is true if the rule holds). Our goal

is to infer the most likely values for a set of unobserved ground predicates, given a set of observed ground predicate values and logic rules.

In *probabilistic logic*, we are instead interested in inferring the probabilities of these ground predicates and rules being true, given a set of observed ground predicates and rules. Furthermore, the truth values of ground predicates and rules may be continuous and lie in the interval $[0, 1]$, instead of being boolean, representing the probability that the corresponding ground predicate or rule is true. In this case, boolean logic operators, such as AND ($\wedge$), OR ($\vee$), NOT ($\neg$), and IMPLIES ($\rightarrow$), need to be redefined. For the next section, we will assume their classical logical interpretation.

## 3.2 Model

As described earlier, our goal is to estimate the true accuracies of each of the function approximations, $\hat{f}_1^d, \ldots, \hat{f}_{N^d}^d$ for $d = 1, \ldots, D$, using only unlabeled data, as well as the response of the true underlying functions, $f^d$. We now define the logic rules that we perform inference over in order to achieve that goal. The rules are defined in terms of the following predicates, for $d = 1, \ldots, D$:

- Function Approximation Outputs: $\hat{f}_j^d(X)$, defined over all approximations $j = 1, \ldots, N^d$, and inputs $X \in \mathcal{X}$, for which the corresponding function approximation has provided a response. Note that the values of these ground predicates lie in $[0, 1]$ due to their probabilistic nature (i.e., they do not have to be binary, as in related work), and some of them are observed.
- Target Function Outputs: $f^d(X)$, defined over all inputs $X \in \mathcal{X}$. Note that, in the purely unsupervised setting, none of these ground predicate values are observed, in contrast with the semi-supervised setting.
- Function Approximation Error Rates: $e_j^d$, defined over all approximations $j = 1, \ldots, N^d$. Note that none of these ground predicate values are observed. The primary goal of this paper is to infer their values.

The goal of the logic rules we define is two-fold: (i) to combine the function approximation outputs in a single output value, and (ii) to account for the logical constraints between the domains. We aim to achieve both goals while accounting for the error rates of the function approximations. We first define a set of rules that relate the function approximation outputs with the true underlying function output. We call this set of rules the *ensemble rules* and we describe them in the following section. We then discuss how to account for the logical constraints between the domains.

### 3.2.1 Ensemble Rules

This first set of rules specifies a relation between the target function outputs, $f^d(X)$, and the function approximation outputs, $\hat{f}_j^d(X)$, independent of the logical constraints:

$$\hat{f}_j^d(X) \wedge \neg e_j^d \rightarrow f^d(X), \ \neg \hat{f}_j^d(X) \wedge \neg e_j^d \rightarrow \neg f^d(X), \tag{1}$$

$$\hat{f}_j^d(X) \wedge e_j^d \rightarrow \neg f^d(X), \text{and } \neg \hat{f}_j^d(X) \wedge e_j^d \rightarrow f^d(X), \tag{2}$$

for $d = 1, \ldots, D$, $j = 1, \ldots, N^d$, and $X \in \mathcal{X}$. In words: (i) the first set of rules state that if a function approximation is not making an error, its output should match the output of the target function, and (ii) the second set of rules state that if a function approximation is making an error, its output should not match the output of the target function.

An interesting point to make is that the ensemble rules effectively constitute a weighted majority vote for combining the function approximation outputs, where the weights are determined by the error rates of the approximations. These error rates are implicitly computed based on agreement between the function approximations. This is related to the work of Platanios et al. [2014]. There, the authors try to answer the question of whether consistency in the outputs of the approximations implies correctness. They directly use the agreement rates of the approximations in order to estimate their error rates. Thus, there exists an interesting connection in our work in that we also implicitly use agreement rates to estimate error rates, and our results, even though improving upon theirs significantly, reinforce their claim.

**Identifiability.** Let us consider flipping the values of all error rates (i.e., setting them to one minus their value) and the target function responses. Then, the ensemble logic rules would evaluate to the same value as before (e.g., satisfied or unsatisfied). Therefore, the error rates and the target function values are not identifiable when there are no logical constraints. As we will see in the next

section, the constraints may sometimes help resolve this issue as, often, the corresponding logic rules do not exhibit that kind of symmetry. However, for cases where that symmetry exists, we can resolve it by assuming that most of the function approximations have error rates better than chance (i.e., $< 0.5$). This can be done by considering the two rules: (i) $\hat{f}_j^d(X) \rightarrow f^d(X)$, and $\neg \hat{f}_j^d(X) \rightarrow \neg f^d(X)$, for $d = 1, \ldots, D$, $j = 1, \ldots, N^d$, and $X \in \mathcal{X}$. Note that all that these rules imply is that $\hat{f}_j^d(X) = f^d(X)$ (i.e., they represent the prior belief that function approximations are correct). As will be discussed in section 3.3, in probabilistic frameworks where rules are weighted with a real value in $[0, 1]$, these rules will be given a weight that represents their significance or strength. In such a framework, we can consider using a smaller weight for these prior belief rules, compared to the remainder of the rules, which would simply correspond to a *regularization* weight. This weight can be a tunable or even learnable parameter.

### 3.2.2 Constraints

The space of possible logical constraints is huge; we do not deal with every possible constraint in this paper. Instead, we focus our attention on two types of constraints that are abundant in structured prediction problems in machine learning, and which are motivated by the use of our method in the context of NELL:

- Mutual Exclusion: If domains $d_1$ and $d_2$ are mutually exclusive, then $f^{d_1} = 1$ implies that $f^{d_2} = 0$. For example, in the NELL setting, if a NP belongs to the "city" category, then it cannot also belong to the "animal" category.
- Subsumption: If $d_1$ subsumes $d_2$, then if $f^{d_2} = 1$, we must have that $f^{d_1} = 1$. For example, in the NELL setting, if a NP belongs to the "cat" category, then it must also belong to the "animal" category.

This set of constraints is sufficient to model most ontology constraints between categories in NELL, as well as a big subset of the constraints more generally used in practice.

**Mutual Exclusion Rule.** We first define the predicate $\text{ME}(d_1, d_2)$, indicating that domains $d_1$ and $d_2$ are mutually exclusive[1]. This predicate has value 1 if domains $d_1$ and $d_2$ are mutually exclusive, and value 0 otherwise, and its truth value is observed for all values of $d_1$ and $d_2$. Furthermore, note that it is *symmetric*, meaning that if $\text{ME}(d_1, d_2)$ is true, then $\text{ME}(d_2, d_1)$ is also true. We define the mutual exclusion logic rule as:

$$\text{ME}(d_1, d_2) \wedge \hat{f}_j^{d_1}(X) \wedge f^{d_2}(X) \rightarrow e_j^{d_1}, \tag{3}$$

for $d_1 \neq d_2 = 1, \ldots, D$, $j = 1, \ldots, N^{d_1}$, and $X \in \mathcal{X}$. In words, this rule says that if $f^{d_2}(X) = 1$ and domains $d_1$ and $d_2$ are mutually exclusive, then $\hat{f}_j^{d_1}(X)$ must be equal to 0, as it is an approximation to $f^{d_1}(X)$ and ideally we want that $\hat{f}_j^{d_1}(X) = f^{d_1}(X)$. If that is not the case, then $\hat{f}_j^{d_1}$ must be making an error.

**Subsumption Rule.** We first define the predicate $\text{SUB}(d_1, d_2)$, indicating that domain $d_1$ subsumes domain $d_2$. This predicate has value 1 if domain $d_1$ subsumes domain $d_2$, and 0 otherwise, and its truth value is always observed. Note that, unlike mutual exclusion, this predicate is not symmetric. We define the subsumption logic rule as:

$$\text{SUB}(d_1, d_2) \wedge \neg \hat{f}_j^{d_1}(X) \wedge f^{d_2}(X) \rightarrow e_j^{d_1}, \tag{4}$$

for $d_1, d_2 = 1, \ldots, D$, $j = 1, \ldots, N^{d_1}$, and $X \in \mathcal{X}$. In words, this rule says that if $f^{d_2}(X) = 1$ and $d_1$ subsumes $d_2$, then $\hat{f}_j^{d_1}(X)$ must be equal to 1, as it is an approximation to $f^{d_1}(X)$ and ideally we want that $\hat{f}_j^{d_1}(X) = f^{d_1}(X)$. If that is not the case, then $\hat{f}_j^{d_1}$ must be making an error.

Having defined all of the logic rules that comprise our model, we now describe how to perform inference under such a probabilistic logic model, in the next section. Inference in this case comprises determining the most likely truth values of the unobserved ground predicates, given the observed predicates and the set of rules that comprise our model.

### 3.3 Inference

In section 3.1 we introduced the notion of probabilistic logic and we defined our model in terms of probabilistic predicates and rules. In this section we discuss in more detail the implications of using probabilistic logic, and the way in which we perform inference in our model. There exist various probabilistic logic frameworks, each making different assumptions. In what is arguably the most popular such framework, Markov Logic Networks (MLNs) [Richardson and Domingos, 2006], inference is performed over a constructed Markov Random Field (MRF) based on the model logic rules. Each potential function in the MRF corresponds to a ground rule and takes an arbitrary positive value when the ground rule is satisfied and the value 0 otherwise (the positive values are often called *rule weights* and can be either fixed or learned). Each variable is boolean-valued and corresponds to a ground predicate. MLNs are thus a direct probabilistic extension to boolean logic. It turns out that due to the discrete nature of the variables in MLNs, inference is NP-hard and can thus be very inefficient. Part of our goal in this paper is for our method to be applicable at a very large scale (e.g., for systems like NELL). We thus resorted to Probabilistic Soft Logic (PSL) [Bröcheler et al., 2010], which can be thought of as a convex relaxation of MLNs.

Note that the model proposed in the previous section, which is also the primary contribution of this paper, can be used with various probabilistic logic frameworks. Our choice, which is described in this section, was motivated by scalability. One could just as easily perform inference for our model using MLNs, or any other such framework.

#### 3.3.1 Probabilistic Soft Logic (PSL)

In PSL, models, which are composed of a set of logic rules, are represented using hinge-loss Markov random fields (HL-MRFs) [Bach et al., 2013]. In this case, inference amounts to solving a convex optimization problem. Variables of the HL-MRF correspond to soft truth values of ground predicates. Specifically, a HL-MRF, $f$, is a probability density over $m$ random variables, $\mathbf{Y} = \{Y_1, \ldots, Y_m\}$ with domain $\mathbf{D} = [0,1]^m$, corresponding to the unobserved ground predicate values. Let $\mathbf{X} = \{X_1, \ldots, X_n\}$ be an additional set of variables with known values in the domain $[0,1]^n$, corresponding to observed ground predicate values. Let $\phi = \{\phi_1, \ldots, \phi_k\}$ be a finite set of $k$ continuous potential functions of the form $\phi_j(\mathbf{X}, \mathbf{Y}) = (\max\{\ell_j(\mathbf{X}, \mathbf{Y}), 0\})^{p_j}$, where $\ell_j$ is a linear function of $\mathbf{X}$ and $\mathbf{Y}$, and $p_j \in \{1, 2\}$. We will soon see how these functions relate to the ground rules of the model. Given the above, for a set of non-negative free parameters $\lambda = \{\lambda_1, \ldots, \lambda_k\}$ (i.e., the equivalent of MLN rule weights), the HL-MRF density is defined as:

$$f(\mathbf{Y}) = \frac{1}{Z} \exp - \sum_{j=1}^{k} \lambda_j \phi_j(\mathbf{X}, \mathbf{Y}), \tag{5}$$

where $Z$ is a normalizing constant so that $f$ is a proper probability density function. Our goal is to infer the most probable explanation (MPE), which consists of the values of $\mathbf{Y}$ that maximize the likelihood of our data[2]. This is equivalent to solving the following convex problem:

$$\min_{\mathbf{Y} \in [0,1]^m} \sum_{j=1}^{k} \lambda_j \phi_j(\mathbf{X}, \mathbf{Y}). \tag{6}$$

Each variable $X_i$ or $Y_i$ corresponds to a soft truth value (i.e., $Y_i \in [0,1]$) of a ground predicate. Each function $\ell_j$ corresponds to a measure of the *distance to satisfiability* of a logic rule. The set of rules used is what characterizes a particular PSL model. The rules represent prior knowledge we might have about the problem we are trying to solve. For our model, these rules were defined in section 3.2. As mentioned above, variables are allowed to take values in the interval $[0,1]$. We thus need to define what we mean by the truth value of a rule and its distance to satisfiability. For the logical operators AND ($\wedge$), OR ($\vee$), NOT ($\neg$), and IMPLIES ($\rightarrow$), we use the definitions from Łukasiewicz Logic [Klir and Yuan, 1995]: $P \wedge Q \triangleq \max\{P + Q - 1, 0\}$, $P \vee Q \triangleq \min\{P + Q, 1\}$, $\neg P \triangleq 1 - P$, and $P \rightarrow Q \triangleq \min\{1 - P + Q, 1\}$. Note that these operators are a simple continuous relaxation of the corresponding boolean operators, in that for boolean-valued variables, with 0 corresponding to FALSE and 1 to TRUE, they are equivalent. By writing all logic rules in the form $B_1 \wedge B_2 \wedge \cdots \wedge B_s \rightarrow H_1 \vee H_2 \vee \cdots \vee H_t$, it is easy to observe that the distance to satisfiability

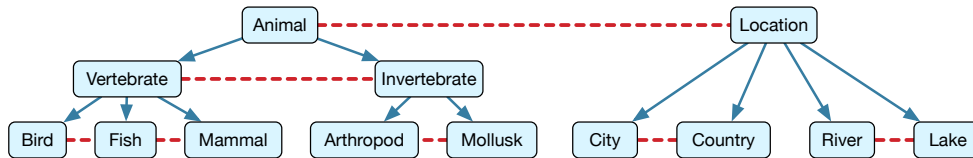

Figure 2: Illustration of the NELL-11 data set constraints. Each box represents a label, each blue arrow represents a subsumption constraint, and each set of labels connected by a red dashed line represents a mutually exclusive set of labels. For example, `Animal` subsumes `Vertebrate` and `Bird`, `Fish`, and `Mammal` are mutually exclusive.

(i.e., 1 minus its truth value) of a rule evaluates to $\max\left\{0, \sum_{i=1}^{s} B_i - \sum_{j=1}^{t} H_t + 1 - s\right\}$. Note that any set of rules of first-order predicate logic can be represented in this form [Bröcheler et al., 2010], and that minimizing this quantity amounts to making the rule "more satisfied".

In order to complete our system description we need to describe: (i) how to obtain a set of ground rules and predicates from a set of logic rules of the form presented in section 3.2 and a set of observed ground predicates, and define the objective function of equation 6, and (ii) how to solve the optimization problem of that equation to obtain the most likely truth values for the unobserved ground predicates. These two steps are described in the following two sections.

### 3.3.2 Grounding

*Grounding* is the process of computing all possible groundings of each logic rule to construct the inference problem variables and the objective function. As already described in section 3.3.1, the variables $\mathbf{X}$ and $\mathbf{Y}$ correspond to ground predicates and the functions $\ell_j$ correspond to ground rules. The easiest way to ground a set of logic rules would be to go through each one and create a ground rule instance of it, for each possible value of its arguments. However, if a rule depends on $n$ variables and each variable can take $m$ possible values, then $m^n$ ground rules would be generated. For example, the mutual exclusion rule of equation 3 depends on $d_1$, $d_2$, $j$, and $X$, meaning that $D^2 \times N^{d_1} \times |X|$ ground rule instances would be generated, where $|X|$ denotes the number of values that $X$ can take. The same applies to predicates; $\hat{f}_j^{d_1}(X)$ would result in $D \times N^{d_1} \times |X|$ ground instances, which would become variables in our optimization problem. This approach would thus result in a huge optimization problem rendering it impractical when dealing with large scale problems such as NELL. The key to scaling up the grounding procedure is to notice that many of the possible ground rules are always satisfied (i.e., have distance to satisfiability equal to 0), irrespective of the values of the unobserved ground predicates that they depend upon. These ground rules would therefore not influence the optimization problem solution and can be safely ignored. Since in our model we are only dealing with a small set of predefined logic rule forms, we devised a heuristic grounding procedure that only generates those ground rules and predicates that may influence the optimization. Our grounding algorithm is shown in the supplementary material and is based on the idea that a ground rule is only useful if the function approximation predicate that appears in its body is observed. It turns out that this approach is orders of magnitude faster than existing state-of-the-art solutions such as the grounding solution used by Niu et al. [2011].

### 3.3.3 Solving the Optimization Problem

For large problems, the objective function of equation 6 will be a sum of potentially millions of terms, each one of which only involving a small set of variables. In PSL, the method used to solve this optimization problem is based on the consensus Alternating Directions Method of Multipliers (ADMM). The approach consists of handling each term in that sum as a separate optimization problem using copies of the corresponding variables, while adding the constraint that all copies of each variable must be equal. This allows for solving the subproblems completely in parallel and is thus scalable. The algorithm is summarized in the supplementary material. More details on this algorithm and on its convergence properties can be found in the latest PSL paper [Bach et al., 2015]. We propose a stochastic variation of this consensus ADMM method that is even more scalable.

During each iteration, instead of solving all subproblems and aggregating their solutions in the consensus variables, we sample $K \ll k$ subproblems to solve. The probability of sampling each

Table 1: Mean absolute deviation (MAD) of the error rate rankings and the error rate estimates (lower MAD is better), and area under the curve (AUC) of the label estimates (higher AUC is better). The best results for each experiment, across all methods, are shown in **bolded** text and the results for our proposed method are highlighted in blue.

| | NELL-7 | | | NELL-11 | | |
|---|---|---|---|---|---|---|
| | $\text{MAD}_{\text{error rank}}$ | $\text{MAD}_{\text{error}}$ | $\text{AUC}_{\text{target}}$ | $\text{MAD}_{\text{error rank}}$ | $\text{MAD}_{\text{error}}$ | $\text{AUC}_{\text{target}}$ |
| MAJ | 7.71 | 0.238 | 0.372 | 7.54 | 0.303 | 0.447 |
| AR-2 | 12.0 | 0.261 | 0.378 | 10.8 | 0.350 | 0.455 |
| AR | 11.4 | 0.260 | 0.374 | 11.1 | 0.350 | 0.477 |
| BEE | 6.00 | 0.231 | 0.314 | 5.69 | 0.291 | 0.368 |
| CBEE | 6.00 | 0.232 | 0.314 | 5.69 | 0.291 | 0.368 |
| HCBEE | 5.03 | 0.229 | 0.452 | 5.14 | 0.324 | 0.462 |
| LEE | **3.71** | **0.152** | **0.508** | **4.77** | **0.180** | **0.615** |

| $\times 10^{-2}$ | uNELL-All | | | uNELL-10% | | |
|---|---|---|---|---|---|---|
| | $\text{MAD}_{\text{error rank}}$ | $\text{MAD}_{\text{error}}$ | $\text{AUC}_{\text{target}}$ | $\text{MAD}_{\text{error rank}}$ | $\text{MAD}_{\text{error}}$ | $\text{AUC}_{\text{target}}$ |
| MAJ | **23.3** | 0.47 | **99.9** | 33.3 | 0.54 | 87.7 |
| GIBBS-SVM | 102.0 | 2.05 | 28.6 | 101.7 | 2.15 | 28.2 |
| GD-SVM | 26.7 | 0.42 | 71.3 | 93.3 | 1.90 | 67.8 |
| DS | 170.0 | 7.08 | 12.1 | 180.0 | 6.96 | 12.3 |
| AR-2 | 48.3 | 2.63 | 96.7 | 50.0 | 2.56 | 96.4 |
| AR | 48.3 | 2.60 | 96.7 | 48.3 | 2.52 | 96.4 |
| BEE | 40.0 | 0.60 | 99.8 | 31.7 | 0.64 | 79.5 |
| CBEE | 40.0 | 0.61 | 99.8 | 118.0 | 45.40 | 55.4 |
| HCBEE | 81.7 | 2.53 | 99.4 | 81.7 | 2.45 | 84.9 |
| LEE | 30.0 | **0.37** | 96.5 | **30.0** | **0.43** | **97.3** |

| $\times 10^{-1}$ | uBRAIN-All | | | uBRAIN-10% | | |
|---|---|---|---|---|---|---|
| | $\text{MAD}_{\text{error rank}}$ | $\text{MAD}_{\text{error}}$ | $\text{AUC}_{\text{target}}$ | $\text{MAD}_{\text{error rank}}$ | $\text{MAD}_{\text{error}}$ | $\text{AUC}_{\text{target}}$ |
| MAJ | 8.76 | 0.57 | 8.49 | 1.52 | 0.68 | 7.84 |
| GIBBS-SVM | 7.77 | 0.43 | 4.65 | 1.51 | 0.66 | 5.28 |
| GD-SVM | **7.60** | 0.44 | 5.24 | 1.50 | 0.68 | 8.56 |
| DS | 7.77 | 0.44 | 8.76 | **1.32** | 0.63 | 4.59 |
| AR-2 | 16.40 | 0.87 | 9.71 | 2.28 | 0.97 | 9.89 |
| BEE | 7.98 | 0.40 | 9.32 | 1.38 | 0.63 | 9.35 |
| CBEE | 10.90 | 0.43 | 9.34 | 1.77 | 0.89 | 9.30 |
| HCBEE | 28.10 | 0.85 | 9.20 | 3.25 | 0.97 | 9.37 |
| LEE | **7.60** | **0.38** | **9.95** | **1.32** | **0.47** | **9.98** |

subproblem is proportional to the distance of its variable copies from the respective consensus variables. The intuition and motivation behind this approach is that at the solution of the optimization problem, all variable copies should be in agreement with the consensus variables. Therefore, prioritizing subproblems whose variables are in greater disagreement with the consensus variables might facilitate faster convergence. Indeed, this modification to the inference algorithm allowed us to apply our method to the NELL data set and obtain results within minutes instead of hours.

## 4 Experiments

Our implementation as well as the experiment data sets are available at `https://github.com/eaplatanios/makina`.

**Data Sets.** First, we considered the following two data sets with logical constraints:

- NELL-7: Classify noun phrases (NPs) as belonging to a category or not (categories correspond to domains in this case). The categories considered for this data set are `Bird`, `Fish`, `Mammal`, `City`, `Country`, `Lake`, and `River`. The only constraint considered is that all these categories are mutually exclusive.
- NELL-11: Perform the same task, but with the categories and constraints illustrated in figure 2.

For both of these data sets, we have a total of 553,940 NPs and 6 classifiers, which act as our function approximations and are described in [Mitchell et al., 2015]. Not all of the classifiers provide a response every input NP. In order to show the applicability of our method in cases where there are no logical constraints between the domains, we also replicated the experiments of Platanios et al. [2014]:

- uNELL: Same task as NELL-7, but without considering the constraints and using 15 categories, 4 classifiers, and about 20,000 NPs per category.

- uBRAIN: Classify which of two 40 second long story passages corresponds to an unlabeled 40 second time series of Functional Magnetic Resonance Imaging (fMRI) neural activity. 11 classifiers were used and the domain in this case is defined by 11 different locations in the brain, for each of which we have 924 examples. Additional details can be found in [Wehbe et al., 2014].

**Methods.** Some of the methods we compare against do not explicitly estimate error rates. Rather, they combine the classifier outputs to produce a single label. For these methods, we produce an estimate of the error rate using these labels and compare against this estimate.

1. Majority Vote (MV): This is the most intuitive method and it consists of taking the most common output among the provided function approximation responses, as the combined output.
2. GIBBS-SVM/GD-SVM: Methods of Tian and Zhu [2015].
3. DS: Method of Dawid and Skene [1979].
4. Agreement Rates (AR): This is the method of Platanios et al. [2014]. It estimates error rates but does not infer the combined label. To that end, we use a weighted majority vote, where the classifiers' predictions are weighted according to their error rates in order to produce a single output label. We also compare against a method denoted by AR-2 in our experiments, which is the same method, except only pairwise function approximation agreements are considered.
5. BEE/CBEE/HCBEE: Methods of Platanios et al. [2016].

In the results, LEE stands for Logic Error Estimation and refers to the proposed method of this paper.

**Evaluation.** We compute the sample error rate estimates using the true target function labels (which are always provided), and we then compute three metrics for each domain and average over domains:

- Error Rank MAD: We rank the function approximations by our estimates and by the sample estimates to produce two vectors with the ranks. We then compute the mean absolute deviation (MAD) between the two vectors, where by MAD we mean the $\ell_1$ norm of the vectors' difference.
- Error MAD: MAD between the vector of our estimates and the vector of the sample estimates, where each vector is indexed by the function approximation index.
- Target AUC: Area under the precision-recall curve for the inferred target function values, relative to the true function values that are observed.

**Results.** First, note that the largest execution time of our method among all data sets was about 10 minutes, using a 2013 15-inch MacBook Pro. The second best performing method, HCBEE, required about 100 minutes. This highlights the scalability of our approach. Results are shown in table 1.

1. NELL-7 and NELL-11 Data Sets: In this case we have logical constraints and thus, this set of results is most relevant to the central research claims in this paper (our method was motivated by the use of such logical constraints). It is clear that our method outperforms all existing methods, including the state-of-the-art, by a significant margin. Both the MADs of the error rate estimation, and the AUCs of the target function response estimation, are significantly better.
2. uNELL and uBRAIN Data Sets: In this case there exist no logical constraints between the domains. Our method still almost always outperforms the competing methods and, more specifically, it always does so in terms of error rate estimation MAD. This set of results makes it clear that our method can also be used effectively in cases where there are no logical constraints.

## Acknowledgements

We would like to thank Abulhair Saparov and Otilia Stretcu for the useful feedback they provided in early versions of this paper. This research was performed during an internship at Microsoft Research, and was also supported in part by NSF under award IIS1250956, and in part by a Presidential Fellowship from Carnegie Mellon University.

## Footnotes

[1]A set of mutually-exclusive domains can be reduced to pairwise ME constraints for all pairs in that set.

[2]As opposed to performing marginal inference which aims to infer the marginal distribution of these values.

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
