[Supplementary Material]

# Supplementary Material for Estimating Accuracy from Unlabeled Data: A Probabilistic Logic Approach

**Emmanouil A. Platanios**
Carnegie Mellon University
Pittsburgh, PA
e.a.platanios@cs.cmu.edu

**Hoifung Poon**
Microsoft Research
Redmond, WA
hoifung@microsoft.com

**Tom M. Mitchell**
Carnegie Mellon University
Pittsburgh, PA
tom.mitchell@cs.cmu.edu

**Eric Horvitz**
Microsoft Research
Redmond, WA
horvitz@microsoft.com

## 1   Grounding Algorithm

---
**Algorithm 1:** Grounding algorithm.

---
**Input:** $\hat{f}_j^d(X)$, for $d = 1, \ldots, D$, and $j = 1, \ldots, N^d$, only for observed values, set of all pairwise mutual-exclusion constraints $ME = \{d_1^i, d_2^i\}_{i=1}^M$, and set of all subsumption constraints $SUB = \{d_1^i, d_2^i\}_{i=1}^S$.

1  Create empty sets $G_p$ and $G_l$
2  **foreach** *observed* $\hat{f}_j^d(X)$ **do**
3      Add $\hat{f}_j^d(X)$, $e_j^d$, and $f^d(X)$ to $G_p$
4      Add $\hat{f}_j^d(X) \wedge \neg e_j^d \rightarrow f^d(X)$ and $\neg \hat{f}_j^d(X) \wedge \neg e_j^d \rightarrow \neg f^d(X)$ to $G_l$
5      Add $\hat{f}_j^d(X) \wedge e_j^d \rightarrow \neg f^d(X)$ and $\neg \hat{f}_j^d(X) \wedge e_j^d \rightarrow f^d(X)$ to $G_l$
6      Add $\hat{f}_j^d(X) \rightarrow f^d(X)$ and $\neg \hat{f}_j^d(X) \rightarrow \neg f^d(X)$ to $G_l$
7      **foreach** *pair* $(d_1, d_2)$ *in ME* **do**
8          **if** $d_1 = d$ **then**
9              Add $f^{d_2}(X)$ to $G_p$
10             Add $ME(d_1, d_2) \wedge \hat{f}_j^{d_1}(X) \wedge f^{d_2}(X) \rightarrow e_j^{d_1}$ to $G_l$
11         **else if** $d_2 = d$ **then**
12             Add $f^{d_1}(X)$ to $G_p$
13             Add $ME(d_2, d_1) \wedge \hat{f}_j^{d_2}(X) \wedge f^{d_1}(X) \rightarrow e_j^{d_2}$ to $G_l$
14     **foreach** *pair* $(d_1, d_2)$ *in SUB* **do**
15         **if** $d_1 = d$ **then**
16             Add $f^{d_2}(X)$ to $G_p$
17             Add $SUB(d_1, d_2) \wedge \neg \hat{f}_j^{d_1}(X) \wedge f^{d_2}(X) \rightarrow e_j^{d_1}$ to $G_l$

---
**Output:** Set of ground predicates $G_p$ and set of ground rules $G_l$.

---

## 2  PSL Consensus ADMM Inference Algorithm

---

**Algorithm 2:** PSL consensus ADMM inference algorithm.

---

**Input:** Observed ground predicate values $\mathbf{X}$, objective terms $\boldsymbol{\ell}$, $\boldsymbol{p}$, rule weights $\boldsymbol{\lambda}$, parameter $\rho$, and mapping from variable copies' indices to consensus variables' indices $\mathcal{G}$.

1   Randomly initialize all $\mathbf{Y}$ (consensus variables) and $\alpha_j$ (Lagrange multipliers) for $j = 1, \ldots, k$, and then randomly initialize the variable copies $\mathbf{y}_j$ for $j = 1, \ldots, k$, corresponding to each subproblem

2   **while** *not converged* **do**

3      **for** $i = 1, \ldots, k$ **do**

4          $\alpha_j \leftarrow \alpha_j + \rho(\mathbf{y}_j - \mathbf{Y}_{\mathcal{G}(j,:)})$

5          $\mathbf{y}_j \leftarrow \arg\min_{\mathbf{y}_j} \left[ \lambda_j [\max\{\ell_j(\mathbf{X}, \mathbf{y}_j)\}]^{p_j} \right.$

6                        $\left. + \frac{\rho}{2} \| \mathbf{y}_j - \mathbf{Y}_{\mathcal{G}(j,:)} + \frac{1}{\rho}\alpha_j \|_2^2 \right]$

7      **for** $i = 1, \ldots, length(\mathbf{Y})$ **do**

8          $\mathbf{Y}_i \leftarrow \frac{\sum_{\mathcal{G}(j,d)=i} \left( [\mathbf{y}_j]_d + \frac{1}{\rho}[\alpha_j]_d \right)}{\sum_{\mathcal{G}(j,d)=i} \mathbf{1}}$

9          Project $\mathbf{Y}_i$ on the interval $[0, 1]$

**Output:** Inferred ground predicate values $\mathbf{Y}$.

---