[Reviews · NeurIPS 2017]

Reviewer 1



This paper builds on the work of Platanios et al. (2014, 2016) on estimating the accuracy of a set of classifiers for a given task using only unlabeled data, based on the agreement behavior of the classifiers. The current work uses a probabilistic soft logic (PSL) model to infer the error rates of the classifiers. The paper also extends this approach to the case where we have multiple related classification tasks: for instance, classifying noun phrases with regard to their membership in multiple categories, some of which subsume others and some of which are mutually exclusive. The paper shows how a PSL model can take into account these constraints among the categories, yielding better error rate estimates and higher joint classification accuracy. [UPDATED] I see this paper as just above the acceptance threshold. It is well written and the methodology seems sound. It provides a compelling example of improving prediction accuracy by applying a non-trivial inference algorithm to multiple variables, not just running feed-forward classification. The paper is less strong in terms of novelty. It's not surprising that a system can achieve better accuracy by using cross-category constraints. The paper does this with an existing PSL system (from Bach et al. 2015). The paper discusses a customized grounding algorithm in Section 3.3.2 (p. 6), but it seems to boil down to iterating over the given instances and classifier predictions and enumerating rule groundings that are not trivially satisfied. The only other algorithmic novelty seems to be the sampling scheme in the second paragraph of Section 3.3.3 (p. 7). [UPDATE: The authors say in their response that this sampling scheme was critical to speeding up inference so it could run at the necessary scale. If this is where a significant amount of the work's novelty lies, it deserves more space and a more thorough exposition.] The paper could also benefit from deeper and more precise analysis. A central choice in the paper is the use of probabilistic soft logic, where variables can take values in [0, 1], rather than a framework such as Markov Logic Networks (MLNs) that deals with discrete variables. The paper says that this choice was a matter of scalability, and "one could just as easily perform inference for our model using MLNs" (lines 213-214, p. 5). I don't think this is true. The proposed model applies logical operators to continuous-valued variables (for the error rates, and the predictions of probabilistic classifiers), which is meaningful in PSL but not MLNs. If one were to model this problem with an MLN, I think one would replace the error rate variables with learnable weights on rules connecting the classifier outputs to the true category values. Such modeling decisions would change the scalability story: there would be no edges in the MLN between variables corresponding to different input instances (e.g., different noun phrases). So one could do exact inference on the sub-networks for the various instances in parallel, and accumulate statistics to use in an EM algorithm to learn the weights. This might be roughly as efficient as the optimization described in Section 3.3.3. [UPDATE: In the MLN I'm envisioning here, with one connected component per noun phrase, information is still shared across instances (noun phrases) during parameter estimation: the weights are shared across all the instances. But when the parameters are fixed, as in the E-step of each EM iteration, the inference algorithm can operate on each noun phrase independently.] The experimental results could also use more analysis. It's not surprising that the proposed LEE algorithm is the best by all metrics in the first row of Table 1 (p. 8), where it takes cross-category constraints into account. I have a number of questions about the other two rows: [UPDATE: Thanks for the answers to these questions.] * What are the "10%" data sets? I didn't find those mentioned in the text. * Does simple majority voting really yield an AUC of 0.999 on the uNELL-All data set? If that's the case, why are the AUCs for majority voting so much lower on NELL-7 and NELL-11? Which setting is most representative of the full problem faced by NELL? * Why does the proposed method yield an AUC of only 0.965 on uNELL-All, performing worse than majority voting and several other methods? [UPDATE: The authors note that the classifier weights from the proposed method end up performing worse than equal weighting on this data set. This could use more analysis. Is there a shortcoming in the method for turning error rates into voting weights?] * On the other three data sets in these rows, the proposed approach does better than the algorithms of Platanios et al. 2014 and 2016, which also use reasonable models. Why does the new approach do better? Is this because the classifiers are probabilistic, and the new approach uses those probabilities while the earlier approaches binarize the predictions? [UPDATE: This still seems to be an open question. The paper would be more satisfying if included some additional analysis on this issue.] * For a given probabilistic classifier on a given labeled data set, the two methods for computing an "error rate" -- one thresholding the prediction at 0.5 and counting errors; the other averaging the probabilities given to incorrect classifications -- will yield different numbers in general. Which way was the "true" error rate computed for these evaluations? Is this fair to all the techniques being evaluated? [UPDATE: The authors clarified that they thresholded the predictions at 0.5 and counted errors, which I agree is fair to both probabilistic and non-probabilistic classifiers.] Smaller questions and comments: * p. 1, Figure 1: If there is a shortage of space, I think this figure could be removed. I skipped over it on my first reading of the paper and didn't feel like I was missing something. * p. 2, line 51: "we define the error rate..." Wouldn't it be helpful to consider separate false positive and false negative rates? * p. 2, line 55: The probabilistic equation for the error rate here doesn't seem syntactically valid. The error rate is not parameterized by a particular instance X, but the equation includes \hat{f}^d_j(X) and (1 - \hat{f}^d_j(X)) outside the P_D terms. Which X is used there? I think what's intended is the average over the data set of the probability that the classifier assigns to the incorrect classification. * p .4, line 148: Were the techniques described in this Identifiability section used in the experiments? If so, how was the prior weight set? * p. 5, Equation 3: Why does the model use \hat{f}^{d_1}_j here, rather than using the unobserved predicate f^{d_1} (which would be symmetrical with f^{d_2}) and relying on constraint propagation? * p. 5, line 219: This line starts using the symbol f for a probability density, which is confusing given the extensive use of f symbols for classifiers and target functions. * p. 6, line 223: What is p_j? Is it always 1 in the proposed model? * p. 6, line 225: How were the lambda parameters set in the experiments? * p. 6, line 257: The use of N^{d_1} in these counting expressions seems imprecise. The expressions include factors of D^2 or D, implying that they're summing over all the target functions in D. So why is the N specific to d_1? * p. 7, line 300: "response every input NP". Missing "for" here. * p. 8, Table 1: The use of different scaling factors (10^-2 and 10^-1) in different sections of the table is confusing. I'd recommend just presenting the numbers without scaling.

Reviewer 2



This paper proposes a method for estimating accuracies of a set of classifiers from unlabelled data using probabilistic logic rules. The method can take constraints into account, in particular stating that labels are mutually exclusive or have a subsumption relationship. It is demonstrated to outperform competing methods on four datasets. The paper is carefully written but dense. There are no theoretical results in the paper, and the experimental results have been crammed into 1.5 page. As a result the paper tries to do too many things and doesn't achieve any of them very well: I couldn't reconstruct the method from the paper, and the experiments seem to show that it works well in some carefully engineered settings with special-purpose metrics, but doesn't do a full strengths and weaknesses analysis (as could have been achieved with artificially generated data). Small point: in Table 1 I'm not sure why the units are different in each of the three main rows (x10^-2, x10^-1).

Reviewer 3



# Summary This work proposes an approach to infer the error rate of a set of classifiers using (only) unlabeled data by leveraging possible logical constraints given by the relationships between the classes. This is achieved by translating these constraints into a probabilistic logic model and inferring a state of the unknown variables (the unknown true classes and error rates of the classifier) that has a high probability. The approach is tested on a noun phrase classification task as well as a FMRI classification task both when the constraints are taken into account and when they are not. In both cases the approach compares favourably with existing approaches both in terms of the approximation of the error rate and the estimation of the target classifier. # Quality Overal, I found the work to be described well and the paper to form a coherent story. After reading the work, however, I was left with an understanding of the method, yet little in the way of understanding the result. Most importantly, I do not think I understand why the method would give useful, (and surprisingly good!) results if the constraints between classes are not present (so in the uNELL and uBRAIN settings in the experiments). The text suggests the authors have a better intuition for this, yet, perhaps due to space constraints they were not able to convey these insights in a way that made me better understand these results. Perhaps they can do so in a revised version or the rebuttal. So, in general, my main concern here is that the paper does an excellent job explaining the "how" of the method, perhaps leaving too little space to explore the "why". # Clarity The paper is very clearly written and already seemed to be thoroughly proofread, for which I want to thank the authors. My only typographic comment has to do with the bolded items in Table 1, which at first confused me because they are not only bold but also red. Also, I can not find what the 10\% in the table refers to. Having said this there were a number of claims in the paper that were not directly clear to me, which I hope the authors can clarify in a revised version. In line 63, it was unclear to me what independence (between which two quantities) you are referring to specifically. In line 119 it would be nice if the reference to the related work is made more specific. For the claim starting on line 139 it is not directly clear to me how this follows from the explanation that precedes it. In equation (1), I suppose it was not very clear to me why the error term in these equations does not depend on the specific instance (X). I understand this makes sense later when in the final model the error rate is inferred to a value that works across all instances, but at this point in the explanation this was not clear to me. # Originality & Significance My second main concern is the scope of the work, since it seems to be tightly integrated in an existing research programme. This is clear both from the experiments, which are done on two datasets, and the references, making it hard to judge the efficacy of the method beyond the specific setting studied here. Perhaps the authors can comment on their thoughts on the general applicability of their methods in other classification problems and when they would expect them to work well, especially when the logical constraints are not present as in the uNELL en uBRAIN examples in the paper.